# Effectiveness and costs associated with a lay counselor–delivered, brief problem-solving mental health intervention for adolescents in urban, low-income schools in India: 12-month outcomes of a randomized controlled trial

Kanika Malik[1,¤,‡], Daniel Michelson[2,‡], Aoife M. Doyle[3], Helen A. Weiss[3], Giulia Greco[4], Rooplata Sahu[1], James E. J.[1], Sonal Mathur[1], Paulomi Sudhir[5], Michael King[6], Pim Cuijpers[7], Bruce Chorpita[8], Christopher G. Fairburn[9], Vikram Patel[1,10,11]*

1 Sangath, New Delhi, India, 2 School of Psychology, University of Sussex, Brighton, United Kingdom, 3 Medical Research Council International Statistics & Epidemiology Group, Faculty of Epidemiology and Population Health, London School of Hygiene & Tropical Medicine, London, United Kingdom, 4 Department of Global Health and Development, Faculty of Public Health and Policy, London School of Hygiene & Tropical Medicine, London, United Kingdom, 5 Department of Clinical Psychology, National Institute of Mental Health and Neuro Sciences, Bengaluru, India, 6 Division of Psychiatry, Faculty of Brain Sciences, University College London, London, United Kingdom, 7 Department of Clinical Psychology, Vrije Universiteit, Amsterdam, the Netherlands, 8 Department of Psychology, University of California, Los Angeles, United States of America, 9 Department of Psychiatry, University of Oxford, Oxford, United Kingdom, 10 Department of Global Health and Social Medicine, Harvard Medical School, Boston, United States of America, 11 Harvard TH Chan School of Public Health, Boston, United States of America

¤ Current address: School of Psychology and Counselling, O.P. Jindal Global University, Haryana, India
‡ KM and DM contributed equally to this work and are joint first authors.
* Vikram_Patel@hms.harvard.edu

## Abstract

### Background

Psychosocial interventions for adolescent mental health problems are effective, but evidence on their longer-term outcomes is scarce, especially in low-resource settings. We report on the 12-month sustained effectiveness and costs of scaling up a lay counselor–delivered, transdiagnostic problem-solving intervention for common adolescent mental health problems in low-income schools in New Delhi, India.

### Methods and findings

Participants in the original trial were 250 school-going adolescents (mean [M] age = 15.61 years, standard deviation [SD] = 1.68), including 174 (69.6%) who identified as male. Participants were recruited from 6 government schools over a period of 4 months (August 20 to December 14, 2018) and were selected on the basis of elevated mental health symptoms and distress/functional impairment. A 2-arm, randomized controlled trial design was used to examine the effectiveness of a lay counselor–delivered, problem-solving intervention (4 to 5 sessions over 3 weeks) with supporting printed booklets (intervention arm) in comparison with problem solving delivered via printed booklets alone (control arm), at the original

**Data Availability Statement:** The 'PRIDE 12-month outcome dataset' that underpins the analysis in this paper are hosted in LSHTM Data Compass at https://doi.org/10.17037/DATA.00002147. In accordance with ethical constraints established when obtaining participant consent, data can only be made available to interested parties on the condition that they sign an agreement stating that they will protect participant confidentiality. To request access, please submit a data request at https://doi.org/10.17037/DATA.00002147.

**Funding:** This research was funded by a Wellcome Trust Principal Research Fellowship grant to VP (106919/Z/15/Z), https://wellcome.ac.uk. The funder had no role in study design, data collection and analysis, decision to publish, or preparation of the manuscript.

**Competing interests:** The authors of this manuscript have read the journal's policy and the authors of this manuscript have the following competing interests: VP is an Academic Editor on PLOS Medicine's editorial board.

**Abbreviations:** AMD, adjusted mean difference; AOR, adjusted odds ratio; CI, confidence interval; CONSORT, Consolidated Standards of Reporting Trials; d, effect size; LMIC, low- and middle-income country; LTFU, lost to follow-up; M, mean; P-O-D, "problem," "options," and "do it"; PRIDE, PRemIum for aDolEscents; PSS, Perceived Stress Scale; QALY, quality-adjusted life year; SAE, serious adverse event; SD, standard deviation; SDQ, Strengths and Difficulties Questionnaire; SWEMWBS, Short Warwick–Edinburgh Mental Well-being Scale; USD, United States dollar; YTP, Youth Top Problems.

endpoints of 6 and 12 weeks. The protocol was modified, as per the recommendation of the Trial Steering Committee, to include a post hoc extension of the follow-up period to 12 months. Primary outcomes were adolescent-reported psychosocial problems (Youth Top Problems [YTP]) and mental health symptoms (Strengths and Difficulties Questionnaire [SDQ] Total Difficulties scale). Other self-reported outcomes included SDQ subscales, perceived stress, well-being, and remission. The sustained effects of the intervention were estimated at the 12-month endpoint and over 12 months (the latter assumed a constant effect across 3 follow-up points) using a linear mixed model for repeated measures and involving complete case analysis. Sensitivity analyses examined the effect of missing data using multiple imputations. Costs were estimated for delivering the intervention during the trial and from modeling a scale-up scenario, using a retrospective ingredients approach. Out of the 250 original trial participants, 176 (70.4%) adolescents participated in the 12-month follow-up assessment. One adverse event was identified during follow-up and deemed unrelated to the intervention. Evidence was found for intervention effects on both SDQ Total Difficulties and YTP at 12 months (YTP: adjusted mean difference [AMD] = −0.75, 95% confidence interval [CI] = −1.47, −0.03, $p = 0.04$; SDQ Total Difficulties: AMD = −1.73, 95% CI = −3.47, 0.02, $p = 0.05$), with stronger effects over 12 months (YTP: AMD = −0.98, 95% CI = −1.51, −0.45, $p < 0.001$; SDQ Total Difficulties: AMD = −1.23, 95% CI = −2.37, −0.09; $p = 0.03$). There was also evidence for intervention effects on internalizing symptoms, impairment, perceived stress, and well-being over 12 months. The intervention effect was stable for most outcomes on sensitivity analyses adjusting for missing data; however, for SDQ Total Difficulties and impairment, the effect was slightly attenuated. The per-student cost of delivering the intervention during the trial was $3 United States dollars (USD; or $158 USD per case) and for scaling up the intervention in the modeled scenario was $4 USD (or $23 USD per case). The scaling up cost accounted for 0.4% of the per-student school budget in New Delhi. The main limitations of the study's methodology were the lack of sample size calculations powered for 12-month follow-up and the absence of cost-effectiveness analyses using the primary outcomes.

## Conclusions

In this study, we observed that a lay counselor–delivered, brief transdiagnostic problem-solving intervention had sustained effects on psychosocial problems and mental health symptoms over the 12-month follow-up period. Scaling up this resource-efficient intervention is an affordable policy goal for improving adolescents' access to mental health care in low-resource settings. The findings need to be interpreted with caution, as this study was a post hoc extension, and thus, the sample size calculations did not take into account the relatively high attrition rate observed during the long-term follow-up.

## Trial registration

ClinicalTrials.gov NCT03630471.

## Author summary

### Why was this study done?

- The PRemIum for aDolEscents (PRIDE) is a research program that aims to develop a transdiagnostic, stepped care intervention model to address common adolescent mental health problems (anxiety, depression, and conduct difficulties) in low-resource settings. The intervention model comprises a brief problem-solving intervention ("Step 1"), followed by a higher-intensity personalized psychological treatment ("Step 2") for adolescents with persistent problems.

- We previously reported on short-term outcomes from a randomized controlled trial of the first-line problem-solving intervention delivered by lay counselors in secondary schools serving low-income communities in New Delhi, India.

- The current study examined the sustained effectiveness and costs of scaling up the counselor-led problem-solving intervention compared to printed problem-solving materials without counselor input.

### What did the researchers do and find?

- We followed up the original trial participants at 12 months after randomization and collected adolescent-reported outcomes, as well as data on intervention costs using a retrospective ingredients approach.

- The primary analysis showed sustained intervention effects on both psychosocial problems and mental health symptoms.

- The economic analysis showed that the counselor-led problem-solving intervention can be scaled up at a small percentage of the per-student budgetary allocation in government-run schools in New Delhi, India.

### What do these findings mean?

- Despite its brevity and delivery by lay counselors, the problem-solving intervention showed sustained effectiveness.

- Scaling up this low-cost intervention represents an affordable policy goal for improving access to school-based mental health care for adolescents in India and, potentially, in other low-resource settings.

## Introduction

Psychosocial interventions have been advocated to address the growing burden of adolescent mental health problems globally [1,2]. Although there is a large body of research on the short-term outcomes of psychosocial interventions for adolescents with common mental health problems, evidence for their sustained effectiveness is relatively scarce. A systematic review of

youth psychotherapy trials for internalizing and externalizing problems found that only one-third of adolescent-focused trials (52 out of 155) reported longer-term follow-up outcomes, averaging 11 months in duration. The pooled effect size in these follow-up studies was 0.28, indicating a small sustained effect of psychotherapies across diverse problem types [3]. Examination of trial characteristics indicated that most intervention protocols were time intensive (16.5 sessions on average) and involved delivery by specialists. Relatively few studies (*n* = 13) examined interventions targeting comorbid presentations, which represent the majority of "real-world" case mix [3]. Moreover, no longer-term follow-up studies were identified from low- and middle-income countries (LMICs). These regions account for 90% of the global adolescent population [4], yet command a small fraction of mental health service resources and associated research infrastructure worldwide.

The PRemIum for aDolEscents (PRIDE) research program was conceived to meet the need for contextually sensitive, evidence-based interventions that target common adolescent mental health problems in India and LMICs more generally. The PRIDE intervention model is situated in secondary schools and built around a transdiagnostic stepped care architecture, which comprises a brief problem-solving intervention ("Step 1") and a higher-intensity personalized psychological treatment ("Step 2") for adolescents who do not respond to the first-line intervention [5,6]. The first step was evaluated in a randomized controlled trial in New Delhi, India, where lay counselors functioned as the delivery agent in the intervention arm [7]. When compared against a control condition consisting of printed problem-solving booklets without counselor input, there was evidence of an intervention effect on one of the 2 primary outcomes (self-reported psychosocial problems) at 6 weeks and 12 weeks, but not on the other primary outcome (self-reported mental health symptoms). However, symptom trajectories suggested that differences between the intervention and control arms were widening at 12 weeks. It was speculated that a longer follow-up period might reveal a distal effect on mental health, as newly learned coping skills were practiced and consolidated over time [7]. The current study, therefore, examined trajectories of trial outcomes for both arms over 12 months following randomization. We also estimated the incremental costs of setting up and delivering the intervention during the trial and used these estimates to enumerate the costs of scaling up the intervention across public schools in New Delhi.

## Methods

### Study design and participants

The original trial protocol (ClinicalTrials.gov NCT03630471) was restricted to short-term endpoints at 6 and 12 weeks. Based on recommendations of the Trial Steering Committee in a meeting on May 20, 2019, a protocol modification was made to conduct an additional follow-up at 12 months after randomization. Ethical approval for the protocol modification was obtained from the Institutional Review Boards of Harvard Medical School (sponsor) and Sangath (implementing organization in India). The protocol and analysis plan for the 12-month follow-up study have been provided as a Supporting information file (S1 Protocol). This study is reported as per the Consolidated Standards of Reporting Trials (CONSORT) guideline (S1 Checklist).

Detailed descriptions of participant recruitment, trial design, and conduct are available elsewhere [8]. The original recruitment was conducted from August 20 to December 4, 2018 in 6 government-run schools (3 all-boys schools, 2 all-girls schools, and 1 coeducational school), catering to low-income communities in New Delhi. Referrals into the trial were generated through whole-school and classroom-level sensitization activities that were intended to (i) raise awareness of available mental health support, and (ii) address factors such as low mental

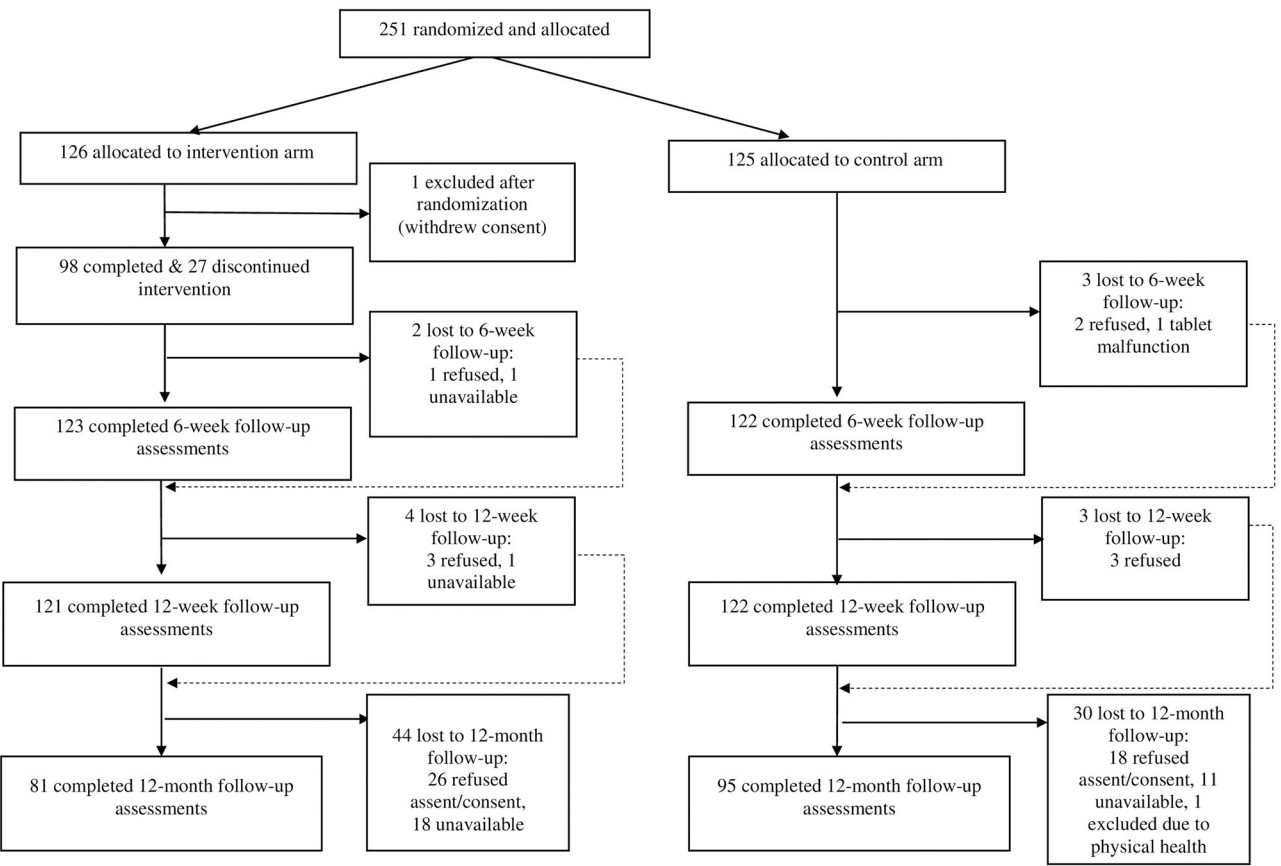

**Fig 1. CONSORT diagram for trial participants.** CONSORT, Consolidated Standards of Reporting Trials.

health literacy and confidentiality concerns that might otherwise limit the demand for school mental health services. The participants were adolescents in grades 9 to 12 who scored at or above the locally validated "borderline" cutoff scores of 19 for boys and 20 for girls on the Strengths and Difficulties Questionnaire (SDQ) Total Difficulties scale [9,10], reported an "abnormal" score of 2 or more on the SDQ Impact scale [9], and indicated persistent problems of more than 1 month on the SDQ Chronicity index. Adolescents were excluded if they needed urgent medical attention from a specialist, were receiving another mental health intervention, had taken part in previous PRIDE studies, demonstrated receptive/expressive language difficulties that affected their ability to participate fully in trial procedures, or declined consent for research participation.

The estimated sample size for the originally stipulated primary endpoint assumed loss to follow-up of 15% at 6 weeks (based on piloting) and a 1:1 allocation ratio to provide over 90% power to detect an effect size of 0.5 (and 80% power to detect an effect size of 0.4) at the α level of 0.025 for each of the 2 primary outcomes [8]. Based on these estimates, a total of 251 participants were initially enrolled, and 125 were randomly allocated to each arm after accounting for one participant in the intervention arm who withdrew consent after randomization (Fig 1). Participants were randomized using the randomization list generated by the senior trial statistician and concealed using sequentially numbered sealed envelopes [8]. Baseline demographic and clinical characteristic for the trial participants in the 2 arms are given in Table 1. For the 12-month follow-up, we approached all 250 participants who opted to be contacted about

**Table 1. Baseline characteristics of participants[1].**

| | Control arm ($n$ = 125) | Intervention arm ($n$ = 125) |
|---|---|---|
| **Age** (in years), M (SD) | 15.59 (1.68) | 15.63 (1.68) |
| **Sex**, $n$ (%) | | |
| Females | 38 (30.4%) | 38 (30.4%) |
| Males | 87 (69.6%) | 87 (69.6%) |
| **School grade**, $n$ (%) | | |
| 9th class | 52 (41.6%) | 58 (46.4%) |
| 10th class | 36 (28.8%) | 31 (24.8%) |
| 11th class | 8 (6.4%) | 8 (6.4%) |
| 12th class | 29 (23.2%) | 28 (22.4%) |
| **Primary caregiver age** (in years)[2], M (SD) | 38.30 (8.16) | 35.47 (8.81) |
| **Primary caregiver education**[2], $n$ (%) | | |
| No formal education | 25 (21.2%) | 25 (21.2%) |
| Completed primary | 3 (2.5%) | 2 (1.7%) |
| Completed secondary school | 56 (47.5%) | 57 (48.3%) |
| Completed higher secondary and above | 24 (20.3%) | 27 (22.9%) |
| Not available | 10 (8.5%) | 7 (5.9%) |
| **Primary caregiver occupation**[2], $n$ (%) | | |
| Not employed outside home | 45 (38.1%) | 46 (39.0%) |
| Manual | 59 (50.0%) | 52 (44.1%) |
| Clerical | 9 (7.6%) | 5 (4.2%) |
| Professional | 3 (2.5%) | 4 (3.4%) |
| Other | 2 (1.7%) | 11 (9.3%) |
| **SDQ Total Difficulties scale**[3], M (SD) | 23.12 (3.01) | 23.22 (3.31) |
| **SDQ Impact scale**[3], M (SD) | 5.20 (2.37) | 5.38 (2.41) |
| **SDQ Internalizing symptoms subscale**[3], M (SD) | 11.96 (2.45) | 12.09 (2.49) |
| **SDQ Externalizing symptoms subscale**[3], M (SD) | 11.16 (2.40) | 11.14 (2.37) |
| **SDQ Chronicity index**[3], $n$ (%) | | |
| 1 to 5 months | 41 (33.1%) | 35 (28.0%) |
| 6 to 12 months | 17 (13.7%) | 22 (17.6%) |
| Over 1 year | 66 (53.2%) | 68 (54.4%) |
| **YTP score**, M (SD) | 7.35 (2.06) | 7.24 (2.24) |
| **PSS-4 score**, M (SD) | 9.22 (2.47) | 9.04 (2.49) |
| **SWEMWBS score**, M (SD) | 20.92 (5.28) | 20.50 (4.82) |

[1] Denominator for each cell is number of participants in that arm, until otherwise specified.

[2] Primary caregiver data not collected for 14 participants (7 in each arm) as index adolescent aged above 18 years; denominator for each row under this head is 118.

[3] Baseline SDQ details were missing for one participant in the control arm; see Michelson and colleagues [7].

M, mean; PSS-4, Perceived Stress Scale 4-item version; SD, standard deviation; SDQ, Strengths and Difficulties Questionnaire; SWEMWBS, Short Warwick–Edinburgh Mental Well-Being Scale; YTP, Youth Top Problems.

future research during the original consent procedure. One week prior to the 12-month follow-up date, attempts were made to establish telephone contact with the original trial participants in order to gauge their interest and availability for participation in the long-term follow-up study. Contact was also made with parents/guardians ("caregivers") for adolescents aged less than 18 years. If agreed, a face-to-face appointment was scheduled at the individual's

home or in school, as per their preference. Written assent (or consent for individuals aged 18 years or older) was obtained from adolescents, followed by consent from a caregiver for adolescents aged less than 18 years. Those providing assent and consent completed the self-reported outcome measures in Hindi on a digital tablet device at a convenient location. In contrast to 6- and 12-week follow-up assessments, no follow-up data were collected from caregivers at 12 months. The 12-month follow-up assessments were completed between August 26, 2019 and January 2, 2020.

## Intervention and control arms

Participants in the intervention arm received 4 to 5 individual face-to-face sessions of a problem-solving intervention, spread over 3 weeks, and delivered on school premises. The frequency of sessions was flexible according to the preference and availability of individual participants. Participants attended sessions during a free period at school or else were excused from a scheduled class. Problem solving was introduced and practiced using a 3-step heuristic (P-O-D, referring to "problem," "options," and "do it"), and participants were supported to put this into practice for one or more prioritized problems. The final session focused on consolidating and discussing use of problem-solving skills in different situations. Additionally, participants received a set of 3 illustrated booklets. These booklets contained contextually appropriate stories to explain the problem-solving steps and also included suggestions for home-based practice exercises. The booklets were provided sequentially over 3 sessions, and a poster summarizing the problem-solving steps was distributed in the final session. The intervention was delivered by 8 Hindi speaking, college graduates, who had no formal clinical qualifications or prior experience of delivering a psychological treatment. These counselors were employed by the implementing organization (Sangath) and trained through a combination of office-based learning and supervised field practice. Weekly supervision was provided in a peer group format, which was moderated by a rotating counselor and overseen by a masters- or doctoral-level psychologist. Audio-taped sessions (1 or 2 per week) were reviewed and rated on a number of key competencies with corrective feedback provided as appropriate; further details are described elsewhere [7].

Process indicators from the intervention arm indicated that a mean (M) of 4.06 (standard deviation [SD] = 1.6) sessions were delivered per participant. The average session duration was 23.27 minutes (SD = 4.3). A total of 98 (78.4%) participants completed the intervention (i.e., attended 4 or more sessions) [7]. Reasons for noncompletion for the 27 participants were documented by counselors using a checklist with prespecified categories. These included rapid resolution of the identified problems ($n$ = 13, 48.1%), competing demands at school ($n$ = 6, 22.2%), persistent absence from school ($n$ = 4, 14.8%), and unspecified ($n$ = 4, 14.8%) (Fig 1).

Control arm participants received the same printed materials as intervention arm participants, but these were handed out by a researcher. The researcher introduced the materials using a brief standardized script, lasting approximately 90 seconds, which explained their purpose and instructions for use. No further guidance or support was provided.

The original trial protocol [8] included a provision for adolescents to receive a more intensive intervention from a psychologist if they continued to experience symptoms or impairment (defined as scoring above thresholds on the SDQ Total Difficulties and/or SDQ Impact scales) at the originally stipulated 12-week endpoint. A total of 16 out of 119 adolescents with persistent difficulties took this up: 9 (16.1%) participants from the intervention arm and 7 (11.1%) participants from the control arm. These participants were not excluded from the 12-month follow-up nor from the complete case analysis. None of the other participants reported receiving any additional mental health intervention.

## Outcome measures

A detailed description of the adolescent-reported outcome measures is available in the original trial protocol [8]. Two primary outcomes were assessed using self-report measures at 12 months after randomization: (i) mental health symptoms measured using the SDQ Total Difficulties score (range: 0 to 40) [9], which is derived by summing 20 items covering both internalizing and externalizing symptoms; and (ii) psychosocial problems, measured using the Youth Top Problems (YTP) score (range: 0 to 10) [11], which is an idiographic measure for which an overall score is derived by averaging individual ratings for up to 3 prioritized problems nominated by the respondent. Higher scores indicated greater severity of symptoms and psychosocial problems, respectively. As indicated in Michelson and colleagues [7], 2 primary outcomes were included in the trial as we considered that problem solving would lead to reductions in both prioritized problems (an person-centered outcome) and mental health symptoms (a standardized outcome). Self-reported outcomes were also collected for (i) distress/functional impairment (SDQ Impact scale, range: 0 to 10) [9]; (ii) internalizing symptoms (SDQ Internalizing symptoms subscale, range: 0 to 20) [9]; (iii) externalizing symptoms (SDQ Externalizing symptoms subscale, range: 0 to 20) [9]; (iv) perceived stress (Perceived Stress Scale 4-item version [PSS-4], range: 0 to 16) [12]; (v) well-being (Short Warwick–Edinburgh Mental Well-being Scale [SWEMWBS], range: 7 to 35) [13]; and (vi) proportion of remitted cases (assessed using the "crossing clinical threshold method" [14] and defined as scoring below eligibility thresholds on the SDQ Total Difficulties scale and SDQ Impact scale [8]). Additionally, we used a 4-point Likert scale to collect data on adolescents' self-reported use of problem-solving materials ("In the past year, how often have you used the booklets received during the program?") and problem-solving skills in the preceding 12 months ("In the past year, how often have you used the skills learned during the program?").

## Costing framework

A retrospective ingredients approach [15] was used to develop an intervention costing framework based on information obtained from the provider perspective after completing the trial. This included retrospectively mapping various activities, resources, and time allocation used in setting up and implementing the intervention during the trial, with costing based on records provided by the implementing organization, Sangath. Activities included training and recruitment of counselors, referral generation activities in schools, delivery of sessions, and counselors' supervision. Resources covered the materials used for delivering the intervention (booklets, posters, referral forms, manuals, session and supervision record forms, voice recorders, "drop boxes" for collecting paper-based referrals, and laptops and projectors for classroom sensitization), as well as resources used for setting up consulting areas in schools (chairs, tables, and screens). Interviews with staff from Sangath were conducted to assess the percentage of staff time devoted to trial activities.

## Statistical analysis

**Effectiveness analysis.** The statistical analysis plan was finalized before analysis (included in S1 Protocol). Analyses were done with Stata version 15.1. Baseline characteristics were split by follow-up status (i.e., completed or lost to follow-up [LTFU]). These were summarized using descriptive statistics and compared using *t* tests or chi-squared tests, as appropriate. Serious adverse events (SAEs) were reported as the number of individuals that incurred SAEs in each arm, using criteria described in the original trial protocol [8]. Primary outcome analyses at 12-month were conducted using complete case analysis. Analyses were adjusted for the baseline value of each outcome measure, school, and variables associated with missingness at

any time point (class, age, week of enrollment, and baseline YTP M score). The intervention effects on the SDQ and YTP were analyzed using a linear mixed model for repeated measures. This included data from 6 weeks, 12 weeks, and 12 months, with time by group interaction terms to estimate the intervention effect at each time point including at the 12-month endpoint [16]. In addition, for secondary outcome analyses, the intervention effects on the SDQ, YTP, and other self-reported measures were analyzed over 12 months using similar repeated measures analysis. This analysis used data from all 3 follow-up time points (i.e., at 6 weeks, 12 weeks, and 12 months), assuming a constant effect across the time points. Thus, analysis "over 12 months" represents the estimated effect size at each time point and, more generally, throughout the 12 months follow-up period. The assumption of a constant intervention effect over time was tested using a likelihood ratio test comparing this model with one allowing the effect to vary at the 3 time points. Analogous methods using logistic mixed effects regression were used for binary outcomes. Sensitivity analysis examined the effect of missing data using multiple imputations with a linear imputation model (truncated at 0) adjusting for factors associated with missingness for the imputation (age, class, parents' gender, baseline SDQ Total Difficulties and YTP scores, and arm allocation) with 50 imputed datasets. Sensitivity analysis was also used to examine the effect following exclusion of the 16 participants who had received an additional intervention after 12 weeks. Adjusted mean difference (AMD)/adjusted odds ratios (AORs), adjusted effect size (Cohen's d), 95% confidence intervals (CIs), and $p$-values were reported for primary and secondary outcomes.

An exploratory, prespecified effect moderation analysis tested for heterogeneity of intervention effects by the following: baseline chronicity of mental health difficulties, baseline severity of mental health difficulties, YTP typology, and SDQ caseness profile [8]. Post hoc, SDQ caseness profile was removed from moderation analysis, due to the small number of observations (less than 6) in certain cells. In addition, exploratory mediation analyses were conducted to examine whether a priori factor of perceived stress, and 2 process indicators (use of problem-solving materials and use of problem-solving skills) mediated the effects of the intervention on the primary outcomes in the longer term. The 3 variables were examined in separate mediation models using the generalized structural equation models with bootstrapped CIs and the causal steps outlined by Baron and Kenny [17]. A dose–response effect in the intervention arm was examined using mixed effects regression models to assess differences in each of the primary outcomes according to the frequency of session attendance (adjusted as for the primary analyses).

**Cost analysis.** An incremental cost analysis was conducted for setting up and implementing the intervention during the trial. Incremental costs excluded costs that were incurred in both control and intervention arms (i.e., expenses related to printing booklets and referral generation activities). Costs related to evaluation activities and costs incurred during the schools' summer vacation period were additionally excluded. All included costs, including those related to personnel and capital, were allocated based on proportion of use required for setup and implementation. Capital costs were annualized over their expected useful life and discounted at 3%. Costs incurred in Indian rupees were converted to United States dollars (USD) using the annual average exchange rate from the Bank of India and then inflated to 2020 USD using the available figures from the US Labor Department.

Using the same costing framework, we modeled costs of scaling up the lay counselor–delivered problem-solving intervention as a standard implementation program to 20 underresourced schools in New Delhi for an academic year. In modeling the requirements of the hypothetical scale-up, we used findings from the original trial report [7], as related to the number of working days, types of program activities (sensitization, screening, and intervention), the time required for each activity, and time needed for training and supervision. Additionally,

we assumed that each counselor would work full time and engage exclusively in activities related to the problem-solving intervention over the duration of one academic year (not counting holidays). We also assumed that each counselor's available time in school would be fully occupied by referred cases (i.e., there would be no shortfall in demand). Counselor time was costed using the actual cost of employing counselors as per government pay grades [18]. Supervisor time was costed using pay grades of the implementing organization Sangath, which was assumed to provide technical oversight in the scale-up scenario. Detailed assumptions for scale-up are provided in S1 Text of the Supporting information section. Costs related to the development of booklets and training materials were excluded as these had already been developed with the intention of being free to use. Indirect costs, such as space and furniture, were assumed to be already present within schools and were therefore excluded from the scale-up scenario.

## Results

Out of 250 original trial participants, 221 (88.4%) were successfully contacted, and 177 (70.8%) consented to take part in the 12-month assessments. One participant was excluded after consent due to poor health restricting their involvement. Reasons for nonparticipation among the remaining 44 adolescents were lack of interest ($n = 34$, 77.2%), competing time demands at school ($n = 4$, 9.1%), consent declined by parents ($n = 3$, 6.8%), and unspecified ($n = 3$, 6.8%) (Fig 1). The M duration of follow-up was 12.30 months (SD = 0.28), which was on an average 11.56 months (SD = 0.29) after the planned termination of the intervention for those who were "completers" (i.e., attended 4 or 5 sessions). The LTFU rate was higher among participants in the intervention arm (44 out of 125 [35.2%]) than the control arm (30 out of 125 [24.0%]) ($p = 0.07$). LTFU was also higher among participants from grade 12 (25 out of 57 [43.9%]) compared to grade 11 or lower (49 out of 193 [25.4%]) ($p = 0.06$). Participants who were LTFU had higher YTP scores at baseline (M = 7.74, SD = 1.94) than those who completed follow-up (M = 7.11, SD = 2.20) ($p = 0.03$) (S1 Table). One SAE was reported involving a life-threatening event experienced by a research participant. This SAE was determined by the Data and Safety Monitoring Board to be unrelated to the intervention, and the participant's data were retained in the analysis.

### Effectiveness

At the 12-month endpoint, participants in the intervention arm scored, on average, 1.73 points lower on the SDQ Total Difficulties scale than those in the control arm (AMD = −1.73, 95% CI = −3.47, 0.02; d = 0.28, 95% CI = 0.13, 0.43; $p = 0.05$). The average difference in YTP score between the 2 arms at the 12-month endpoint was 0.75 points, in the direction favoring the intervention arm (AMD = −0.75, 95% CI = −1.47, −0.03; d = 0.27, 95% CI = 0.12, 0.42; $p = 0.04$). Assuming a constant intervention effect across 3 follow-up time points, there was strong evidence of an intervention effect on both SDQ Total Difficulties and YTP scores over 12 months (SDQ Total Difficulties: AMD = −1.23, 95% CI = −2.37, −0.09; d = 0.21, 95% CI = 0.05, 0.36; $p = 0.03$; YTP: AMD = −0.98; 95% CI = −1.51, −0.45; d = 0.34, 95% CI = 0.19, 0.50; $p < 0.001$) (Table 2). Although the intervention effects on both primary outcomes increased slightly across the time points (Fig 2), there was no statistical evidence of an interaction between group and time (SDQ Total Difficulties: $p = 0.72$; YTP: $p = 0.51$).

There was evidence of a beneficial intervention effect on other secondary outcomes over 12 months. These included SDQ Impact scores (AMD = −0.51, 95% CI = −0.93, −0.09; d = 0.21, 95% CI = 0.06, 0.36; $p = 0.02$), SDQ Internalizing symptoms subscale score (AMD = −0.76, 95% CI = −1.42, −0.10; d = 0.22, 95% CI = 0.06, 0.37; $p = 0.03$), and PSS-4 score (AMD = −0.54,

**Table 2. Primary and secondary outcomes for 12 months follow-up.**

| Outcomes | At 12 months | | Over 12 months | | AMD/AOR (95% CI) | Adjusted effect size (95% CI) | p-value |
|---|---|---|---|---|---|---|---|
| | Control arm (n = 95)[1] M (SD) | Intervention arm[1] (n = 81) M (SD) | Control arm[2] (n = 339) M (SD) | Intervention arm[2] (n = 325) M (SD) | | | |
| SDQ Total Difficulties scale | 14.14 (6.04) | 13.05 (6.07) | 16.70 (5.99) | 15.78 (5.94) | At 12 months: −1.73 (−3.47, 0.02) | 0.28 (0.13, 0.43) | 0.05 |
| | | | | | Over 12 months: −1.23 (−2.37, −0.09) | 0.21 (0.05, 0.36) | 0.03 |
| YTP | 2.91 (2.65) | 2.22 (2.33) | 3.84 (2.88) | 2.88 (2.63) | At 12 months: −0.75 (−1.47, −0.03) | 0.27 (0.12, 0.42) | 0.04 |
| | | | | | Over 12 months: −0.98 (−1.51, −0.45) | 0.34 (0.19, 0.50) | <0.001 |
| SDQ Impact scale | 1.18 (1.82) | 0.90 (1.77) | 1.87 (2.61) | 1.43 (2.21) | Over 12 months: −0.51 (−0.93, −0.09) | 0.21 (0.06, 0.36) | 0.02[3] |
| SDQ Internalizing subscale | 6.91 (3.45) | 6.25 (3.31) | 8.22 (3.59) | 7.72 (3.35) | Over 12 months: −0.76 (−1.42, −0.10) | 0.22 (0.06, 0.37) | 0.03 |
| SDQ Externalizing subscale | 7.23 (3.44) | 6.80 (3.17) | 8.48 (3.30) | 8.06 (3.26) | Over 12 months: −0.47 (−1.09, 0.14) | 0.14 (0.01, 0.30) | 0.13 |
| PSS-4 | 6.68 (2.45) | 6.23 (2.39) | 7.04 (2.59) | 6.52 (2.47) | Over 12 months: −0.54 (−1.00, −0.08) | 0.21 (0.06, 0.36) | 0.02 |
| SWEMWBS | 25.06 (6.19) | 26.33 (6.61) | 23.74 (5.88) | 24.69 (6.32) | Over 12 months: 1.16 (−0.07, 2.38) | 0.19 (0.04, 0.34) | 0.06 |
| Remitted, n/N (%) | 57/95 (60.64%) | 57/81 (70.37%) | NA | NA | At 12 months: 1.47 (0.73, 2.96) | NA | 0.28 |

[1] This analysis uses outcome data from the 12-month endpoint only.

[2] This analysis uses outcome data from all 3 follow-up time points (i.e., at 6 weeks, 12 weeks, and 12 months), assuming a constant effect across the points.

[3] Estimated using robust standard errors due to heteroskedasticity.

AMD, adjusted mean difference; AOR, adjusted odds ratio; CI, confidence interval; M, mean; PSS-4, Perceived Stress Scale 4-item version; SDQ, Strengths and Difficulties Questionnaire; SWEMWBS, Short Warwick–Edinburgh Mental Well-Being Scale; YTP, Youth Top Problems.

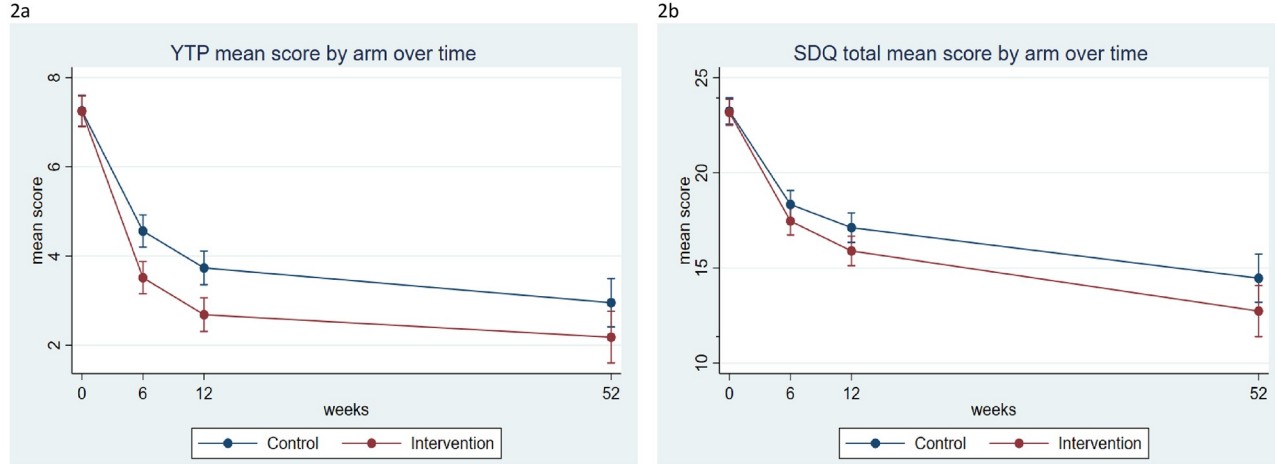

**Fig 2. Primary outcomes over time according to arm. (a)** M YTP score. Error bars indicate 95% CIs. **(b)** M SDQ Total Difficulties score. Error bars indicate 95% CIs. CI, confidence interval; M, Mean; SDQ, Strengths and Difficulties Questionnaire; YTP, Youth Top Problems.

95% CI = −1.00, −0.08; d = 0.21, 95% CI = 0.06, 0.36; *p* = 0.02). A modest effect was observed on the SWEMWBS score, but this did not reach statistical significance (AMD = 1.16, 95% CI = −0.07, 2.38; d = 0.19, 95% CI = 0.04, 0.34; *p* = 0.06). There was no evidence for an intervention effect on the SDQ Externalizing symptoms subscale score (AMD = −0.47, 95% CI = −1.09, 0.14; d = 0.14, 95% CI = 0.01, 0.30; *p* = 0.13) and on remission over 12 months (AMD = 1.47, 95% CI = 0.73, 2.96; *p* = 0.28) (Table 2). The intervention effect on most outcomes was relatively stable across sensitivity analyses, except for the SDQ Total Difficulties and SDQ Impact scores. The effect on these 2 outcomes no longer reached statistical significance in the sensitivity analysis adjusting for missing data, and the effect was slightly attenuated compared with the primary analysis (S2 Table).

## Moderator, mediators, and process indicators

For both primary outcomes, a stronger intervention effect was found for those scoring within the borderline range on the SDQ Total Difficulties scale at baseline, compared with those scoring above the "abnormal" threshold at baseline (SDQ Total Difficulties, borderline range: AMD = −4.27, 95% CI = −7.62, −0.92; SDQ Total Difficulties, abnormal range: AMD = −0.53, 95% CI = −2.55, 1.50; *p* for interaction = 0.06; YTP, borderline range: AMD = −1.79, 95% CI = −3.20, −0.38; YTP, abnormal range: AMD = −0.37, 95% CI = −1.21, 0.48; *p* for interaction = 0.09). However, these differences did not reach statistical significance. There was no evidence for moderation effects by any of the other prespecified variables (baseline chronicity of mental health difficulties, YTP typology, and SDQ caseness profile) on primary outcomes (S3 and S4 Tables).

There was no evidence that perceived stress, use of problem-solving materials or skills at the 12-week endpoint mediated the intervention effect on primary outcomes at 12 months (S5 and S6 Tables).

For the participants in the intervention arm, a dose–response analysis indicated that at 12 months, participants who completed the counselor-led problem-solving intervention had an SDQ Total Difficulties score that was 5.20 points higher than those who discontinued the intervention early (95% CI = 1.72, 8.68, *p* = 0.003). However, intervention completers and noncompleters did not differ on YTP score at 12 months (S7 Table).

## Incremental cost and modeled scale-up cost

Total incremental economic costs for setting up and delivering the counselor-led problem-solving intervention in 6 schools over a period of 22 weeks was $19,742 USD, as compared to the costs of providing booklets alone for the same period. These incremental costs were $3,290 USD per school with per-student incremental costs of $3 USD for a period of 4 months (based on an M of 1,060 students in the targeted classes from each school). The per-case cost was $158 USD when analyzed by number of participants (*n* = 125) in the trial intervention arm. Personnel accounted for 93% of the total costs, whereas capital costs (office and school equipment) and material costs (printed materials and stationery supplies) accounted for 4% and 3% of the total costs, respectively (Table 3).

For the modeled scale-up scenario, our analysis indicated a total cost of $85,920 USD to scale up the counselor-led program in 20 government-run schools in New Delhi over one academic year (39 weeks). This corresponds to a cost of $4,296 USD per school for an academic year and a per-student cost of $4 USD per academic year (Table 3). In the context of the per student per annum expenditure of $890 USD incurred by the local government [19], the per-student cost estimated for scaling up the counselor-led problem-solving intervention is about 0.4% of the budgetary allocation per student. In estimating the maximum caseload that could theoretically be managed by each school counselor within this time frame, we divided the total

**Table 3. Costs for delivering the counselor-led problem-solving intervention in the trial and in a scale-up scenario.**

| | Incremental costs for intervention delivery in 6 schools during the trial (in USD) (Total size of targeted school population = 6,356) | Economic costs for scaling up intervention delivery in 20 schools (in USD) (Total size of targeted school population = 21,200) |
|---|---|---|
| Total cost | 19,742 | 85,920 |
| Total setup cost | 3,638 (18%) | 4,886 (6%) |
| Total implementation cost | 16,104 (82%) | 81,034 (94%) |
| Component-wise costs[1] | | |
| Staff cost | 18,275 (93%) | 63,520 (74%) |
| Materials cost | 658 (3%) | 21,236 (25%) |
| Capital cost | 809 (4%) | 1,164 (1%) |
| School-wise costs | | |
| Per-school cost[2] | 3,290 | 4,296 |
| Per-student cost[3] | 3 | 4 |
| Per-case cost[4] | 158 | 23 |

[1] Component-wise breakdown of total cost into staff, material (e.g., printed sheets), and capital costs (e.g., furniture).

[2] Indicates cost for each school to set up and implement the program in an academic year; calculated by dividing the total cost by the number of schools enrolled in the program.

[3] Indicates per capita cost; calculated by dividing the per-school cost by the average number of students in each school.

[4] Indicates cost for each treated case calculated by dividing the per-school cost by the average number of cases treated in each school.

USD, United States dollar.

working hours available to deliver the intervention by the average number of sessions provided per participant in the main trial. We estimate that in each school, a counselor would be able to provide the intervention for up to 189 eligible students, affording a larger caseload than was possible for each counselor in the trial. This corresponds to 3,780 cases across 20 schools in an academic year with a per-case cost of $23 USD (S1 Text). The higher caseload in the scale-up model led to a higher material costs, accounting for 25% of the total cost, compared to 3% of the total cost during the trial.

## Discussion

We found that a brief problem-solving intervention, delivered by lay counselors and supported by printed booklets, effectively improved adolescents' self-reported psychosocial problems and mental health symptoms at 12 months following randomization, when compared with problem-solving booklets alone. Compared with previously reported postintervention outcomes at 6 and 12 weeks [7], our findings indicated small but sustained intervention effects on psychosocial problems and mental health symptoms over 12 months. However, the intervention effect on mental health symptoms was borderline, which attenuated further in the sensitivity analysis that adjusted for missing data. There was also evidence for longer-term effects on internalizing symptoms, distress/functional impairment, perceived stress, and well-being. Similar to the results obtained at 6 and 12 weeks, we found no evidence of an intervention effect on remission at 12 months. There were modest effects of preintervention mental health symptom severity on both primary outcomes, whereby those scoring above caseness thresholds were more likely to have poorer mental health outcomes and more severe psychosocial problems at 12 months. The observed intervention effect at 12 months was not mediated by any of the hypothesized variables, i.e., perceived stress, use of problem-solving materials, and problem-solving skills.

Most of the existing evidence for the longer-term effectiveness of youth mental health interventions is based on relatively lengthy (16.5 sessions on average), disorder-specific treatment protocols [3]. It is noteworthy that the longer-term effect of this brief, transdiagnostic, problem-solving intervention on mental health symptoms, although small, is comparable to the long-term outcomes achieved by more intensive interventions, mostly delivered by specialists [3,20]. Further, the effect on mental health symptoms was accompanied by a comparable effect on the YTP (an idiographic measure of problems' meaningful to a young person), as well as significant improvements in domains of functional impairment and perceived stress. These long-lasting effects across multiple domains strengthen the evidence base for brief transdiagnostic interventions [21] and indicate the potential for small but meaningful and sustained impact in low-resourced public health contexts [22]. The study also adds to the growing evidence that adequately trained and supervised nonspecialist providers may play an important role in delivering effective psychological interventions for common mental health problems, and this strategy can help to overcome supply side barriers involved in scaling up evidence-based mental health care in low-resource settings [23,24]. However, it is not possible to discern the specific mechanism of the intervention effect in the absence of a fully powered mediation analysis. Qualitative exit interviews with trial participants will be the focus of a future trial report and may shed more light on the potential role of problem-solving skills as an active ingredient in driving therapeutic change, as well as considering the respective contributions of booklets and counselors in the development of problem-solving capacity.

Our dose–response analysis indicated that participants who completed at least 4 intervention sessions reported relatively higher mental health symptoms at 12 months compared with early dropouts from the intervention. A possible explanation for this counterintuitive finding may be related to the observation that early problem resolution was the single most common reason for dropping out from the intervention. In other words, a substantial proportion of participants in the intervention arm appeared to improve rapidly in the space of 1 or 2 weeks, and therefore, opted out of further face-to-face sessions, whereas those with persisting problems continued with the intervention. This raises the prospect of a tailored delivery schedule, where an optimal number of sessions for each participant would be decided through shared decision-making [25]. Another potential area for tailoring is suggested by the observed modest effect of baseline mental health symptom severity on intervention outcomes. This could be taken as support for a stratified care model, where those participants with the most severe symptoms would step up to a more intensive intervention directly, bypassing the low-intensity problem-solving step. Our results require replication in a larger sample before stronger recommendations can be made regarding the utility of progressive or stratified versions of stepped care. Future research should also examine the feasibility and resource implications of dynamic, data-driven clinical decision-making in interventions delivered by nonspecialist providers.

While there is a growing evidence base on the effectiveness of adolescent-focused psychosocial interventions, there is still very little information on the economic costs of effective interventions [26]. Our economic evaluation indicated that a brief, lay counselor–delivered problem-solving intervention could be scaled up at a cost of $4 USD per student (or $23 USD per case), which is a small fraction (0.4%) of the per-student budgetary allocation in government run-schools in New Delhi, India [19]. By hiring full-time school counselors and supporting the costs of their training and supervision, school authorities could provide coverage for adolescents who might not otherwise receive services due to barriers such as transportation, cost, or perceived stigma [27]. In doing so, it is important that the school authorities carefully plan the allocation of counselors' time, as the personnel cost was the biggest cost driver in the per-student cost estimate. Whereas a relatively large school might require more than 1

counselor, several smaller schools could conceivably share 1 full-time counselor to ensure more efficient use of resources. Given these cost variabilities, the per-student costs reported in this study should be considered indicative.

We acknowledge a number of limitations to the study. First, the 12-month follow-up was a post hoc addition to the originally scheduled follow-up assessments at 6 and 12 weeks. Consequently, the sample size calculation did not take into account the numbers expected for the 12-month assessments. The observed LTFU rate at the 12-month point (29.4%) was higher than assumed for the original endpoints. That said, the 12-month LTFU was similar to the attrition rate at long-term follow-up in other randomized trials of interventions for youth anxiety and depression [28,29]. To reduce risk of bias due to attrition while using complete case analysis, variables associated with LTFU were adjusted in these analyses. Further, sensitivity analysis with missing data showed intervention effects were similar to the primary analysis for most variables except SDQ Total Difficulties and Impact scores at 12 months, where the effects were weaker. Second, long-term outcomes may have been affected by participation in other interventions during the extended follow-up period. That said, sensitivity analysis found no difference in effectiveness when removing those participants who were stepped up to a more intensive intervention due to nonresponse at 12 weeks. Third, our analysis was not powered to examine the effects of mediators and moderators. Future research should examine the moderating and mediating roles of baseline characteristics and process variables, which, in turn, can guide the use of stratified models of stepped care and protocols for tailoring intervention dosage. Fourth, the study was not powered to detect an intervention effect on the binary outcome of remission. This may be the reason that we observed effects on most continuous outcomes, yet there was no effect on remission based on crossing prespecified clinical thresholds on the SDQ Total Difficulties and Impact scales. Finally, we did not carry out cost-effectiveness analyses using the primary outcomes, as it would have been difficult to interpret per-unit change for primary outcomes in the absence of a measure of quality-adjusted life years (QALYs), which is most often used in other economic evaluations. Future research could examine statistical mapping algorithms using the responses of condition-specific instruments for the estimation of QALYs. Sensitivity analyses are also needed that modify the assumptions of the delivery parameters in the scale-up scenario, in order to test the robustness of the observed results.

Notwithstanding these limitations, the study demonstrated that a brief problem-solving intervention delivered by lay counselors had durable effects on mental health symptoms, psychosocial problems, and other self-reported outcomes over 12 months. The external validity of findings is strengthened by including adolescents with diverse mental health presentations that reflect real-world case mix. The trial was conducted in government-run schools in low-income, urban areas, which are widespread in India and other LMICs, strengthening the generalizability of findings. In our estimation of costs for scaling up, we have used data on the actual cost of employing personnel and printing materials, which forms the bulk of the cost of providing the intervention. Further research utilizing more sophisticated economic modeling techniques is required to examine possible costs under different scenarios for large-scale implementation.

In conclusion, our findings indicate that a brief problem-solving intervention delivered by nonspecialist school counselors and supported by printed booklets had sustained effects over 12 months and represented good value for money. In view of these longer-term effects, problem-solving delivered by lay counselors can be considered a leading candidate for a low-cost, transdiagnostic intervention. Future evaluations are needed to examine whether incremental benefits can be achieved by supplementing a first-line problem-solving intervention with a more intensive treatment as part of a stepped care model.

## Supporting information

**S1 Protocol. Protocol and analysis plan for the 12-month follow-up.**
(PDF)

**S1 Checklist. CONSORT checklist.** CONSORT, Consolidated Standards of Reporting Trials.
(DOC)

**S1 Text. Assumptions guiding cost estimates for the modeled scale-up of the counselor-led problem-solving intervention in 20 schools for one academic year.**
(DOCX)

**S1 Table. Baseline characteristics of participants who completed follow-up (*n* = 174) and those lost to follow-up (*n* = 76) at 12 months.**
(DOCX)

**S2 Table. Sensitivity analysis using imputed data for primary and secondary outcomes.**
(DOCX)

**S3 Table. Primary outcome (SDQ) by potential effect modifiers at 12 months.** SDQ, Strengths and Difficulties Questionnaire.
(DOCX)

**S4 Table. Primary outcome (YTP) by potential effect modifiers at 12 months.** YTP, Youth Top Problems.
(DOCX)

**S5 Table. Mediation effect of perceived stress, use of problem-solving materials, and problem-solving skills on SDQ Total Difficulties score at 12 months.** SDQ, Strengths and Difficulties Questionnaire.
(DOCX)

**S6 Table. Mediation effect of perceived stress, use of printed materials, and problem-solving skills on YTP at 12 months.** YTP, Youth Top Problems.
(DOCX)

**S7 Table. Dose–response effect on primary outcomes at 12 months for intervention completers and noncompleters.**
(DOCX)

## Acknowledgments

We acknowledge the contributions of study participants, their parents and guardians, school staff, research staff, and counselors who made this work possible. We also acknowledge the team members who contributed to intervention and research activities across the various phases including Bhargav Bhat, Bhagwant Chilhate, Deepak Jangra, Madhuri Krishna, Rachana Parikh, Rhea Sharma, and Sachin Shinde. Finally, we acknowledge the oversight provided by the Trial Steering Committee and the Data and Safety Monitoring Board.

## Author Contributions

**Conceptualization:** Daniel Michelson, Helen A. Weiss, Paulomi Sudhir, Michael King, Pim Cuijpers, Bruce Chorpita, Christopher G. Fairburn, Vikram Patel.

**Data curation:** Aoife M. Doyle, Helen A. Weiss, James E. J.

**Formal analysis:** Kanika Malik, Aoife M. Doyle, Helen A. Weiss, Giulia Greco.

**Funding acquisition:** Vikram Patel.

**Investigation:** Kanika Malik, Daniel Michelson, Rooplata Sahu, Sonal Mathur, Christopher G. Fairburn, Vikram Patel.

**Methodology:** Kanika Malik, Daniel Michelson, Aoife M. Doyle, Helen A. Weiss, Vikram Patel.

**Project administration:** Kanika Malik, Rooplata Sahu.

**Resources:** Vikram Patel.

**Software:** Helen A. Weiss.

**Supervision:** Daniel Michelson, Vikram Patel.

**Visualization:** Aoife M. Doyle, Helen A. Weiss.

**Writing – original draft:** Kanika Malik, Daniel Michelson.

**Writing – review & editing:** Daniel Michelson, Aoife M. Doyle, Helen A. Weiss, Giulia Greco, Rooplata Sahu, James E. J., Sonal Mathur, Paulomi Sudhir, Michael King, Pim Cuijpers, Bruce Chorpita, Christopher G. Fairburn, Vikram Patel.

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
