## [Editor Report · Decision Letter 0]

20 Oct 2020

Dear Dr Patel, 

Thank you for submitting your manuscript entitled "Sustained effectiveness and costs of a lay counsellor-delivered, brief problem-solving intervention for adolescent mental health problems in urban, low-income schools in India: 12-month outcomes of a randomized control trial" for consideration by PLOS Medicine.

Your manuscript has now been evaluated by the PLOS Medicine editorial staff as well as by the Special Issue guest editors and I am writing to let you know that we would like to send your submission out for external peer review.

Kind regards,

Artur A. Arikainen,

Associate Editor

PLOS Medicine

---

## [Decision Letter · Decision Letter 1]

1 Mar 2021

Dear Dr. Patel,

Thank you very much for submitting your manuscript "Sustained effectiveness and costs of a lay counsellor-delivered, brief problem-solving intervention for adolescent mental health problems in urban, low-income schools in India: 12-month outcomes of a randomized control trial" (PMEDICINE-D-20-05026R1) for consideration in PLOS Medicine’s Special Issue on Global Child Health.

Your paper was evaluated by a senior editor and discussed among all the editors here. It was also discussed with an academic editor with relevant expertise, and sent to three independent reviewers, including a statistical reviewer. The reviews are appended at the bottom of this email and any accompanying reviewer attachments can be seen via the link below:

[LINK]

In light of these reviews, I am afraid that we will not be able to accept the manuscript for publication in the journal in its current form, but we would like to consider a revised version that addresses the reviewers' and editors' comments. Obviously we cannot make any decision about publication until we have seen the revised manuscript and your response, and we plan to seek re-review by one or more of the reviewers.

In revising the manuscript for further consideration, your revisions should address the specific points made by each reviewer and the editors. In particular, reviewer 3 has raised critical points pertaining to the study design and analyses. Please also check the guidelines for revised papers at http://journals.plos.org/plosmedicine/s/revising-your-manuscript for any that apply to your paper. In your rebuttal letter you should indicate your response to the reviewers' and editors' comments, the changes you have made in the manuscript, and include either an excerpt of the revised text or the location (eg: page and line number) where each change can be found. Please submit a clean version of the paper as the main article file; a version with changes marked should be uploaded as a marked up manuscript.

We expect to receive your revised manuscript by Mar 22 2021 11:59PM. Please email us (plosmedicine@plos.org) if you have any questions or concerns.

We look forward to receiving your revised manuscript. 

Sincerely,

Caitlin Moyer, PhD 

Associate Editor 

PLOS Medicine

plosmedicine.org

1. Data availability statement: Please revise the data statement. PLOS Medicine requires that the de-identified data underlying the specific results in a published article be made available, without restrictions on access, in a public repository or as Supporting Information at the time of article publication, provided it is legal and ethical to do so. Please see the policy at

http://journals.plos.org/plosmedicine/s/data-availability

and FAQs at

http://journals.plos.org/plosmedicine/s/data-availability#loc-faqs-for-data-policy

Specifically, access to data cannot be dependent on requests to one of the authors of the study. Please update the statement, noting how data underlying the study will be made available.

2. Competing Interests: Please add this statement to the manuscript's Competing Interests: "VP is an Academic Editor on PLOS Medicine's editorial board."

3. Abstract: Methods and Findings: At line 47-48, please indicate you are reporting mean age of the participants. Please provide some demographic background on the adolescent participants, beyond age, such as the population and setting, years during which the study took place.

4. Abstract: Methods and Findings: The retrospective ingredients approach taken could be mentioned.

5. Abstract: Methods and Findings: For the YTP and SDQ main outcomes, please quantify the main results with both 95% CIs and p values.

6. Abstract: Conclusions: We suggest beginning the first sentence with “In this study, we observed…” or similar.

7. Abstract and Author Summary: Where you mention that effects were “moderated by the pre-intervention severity of mental health problems.” could you be more specific in terms of the moderation so the direction/relationship is clear.

8. Methods: While described elsewhere (in ref 8) it would be helpful if some details could be briefly mentioned, such as: briefly summary of details of recruitment and randomization, and background on the lay-counsellors, some details of a definition for the “remission” outcome.

9. Methods: Line 244: Thank you for your note that the statistical analysis plan is available upon request. Please include this as a supporting information file. In addition, you reference the published protocol- however this does not seem to include the updated 12 month time point. It would be helpful if you could please include the modified protocol as a supporting information file.

10. Results: Lines 355-356: Please indicate in the text that the effect on SWEMWBS score did not reach statistical significance.

11. Results: Lines 364-371: It seems as if there were no significant interactions found between SDQ or YTP outcomes and baseline SDQ. Please make this clear, and also present the evidence supporting that there were differences between borderline/abnormal range SDQ at baseline and both SDQ and YTP outcomes (direct comparison, in support of your statement, with the acknowledgement that there is no significant interaction effect).

12. Results: Line 370: Please re-iterate the pre-specified variables here, and please clarify that you are still discussing moderation effects on the primary outcomes.

13. CONSORT Checklist: Please complete the CONSORT checklist and ensure that all components of CONSORT are present in the manuscript. Please add the following statement, or similar, to the Methods: "This study is reported as per the Consolidated Standards of Reporting Trials (CONSORT) guideline (S1 Checklist)."

14. Table 1 and Table 2: Please define all abbreviations used in the table within the legend.

15. S4 Table and S5 Table: Please provide the p-values for all the intervention effects (only provided for Baseline Severity- Borderline row of Table S4).

16. S8 Table: We would suggest using adherence or “adherers” or similar rather than using compliance/compliers.

Comments from the reviewers:

Reviewer #1: This trial makes a very significant and, to my knowledge, original contribution to the literature. It assesses the longer-term value of a problem solving intervention based in a school context and assesses its economic viability for scaling up. The authors are world leading experts in this area and the quality of the manuscript certainly reflects this standing in the field. The results are impressive, important, and worthy of worldwide dissemination. 

I have some comments on the manuscript and suggestions for revisions/consideration. 

Title: The comparator condition isn't mentioned in the title (as per PICOS), and I think it would be helpful to consider indicating it in some form, as the nature of the experimental comparison only becomes clear in the abstract. 

p3, line 47: It may be helpful to state 'years old' after 15.61, so that it is clear what this statistic refers to. 

I was surprised to read that there were two primary outcomes - my understanding of having a primary outcome is that it is, by definition, singular, such that the problems of multiple testing are reduced. This may be a common procedure but I was, as someone relatively familiar with trial design, unfamiliar with it. 

It may be useful to state in the abstract that this was an intention to treat analysis.

From the abstract onwards (e.g., also on page 15), I was a bit confused by the difference between an analysis of outcomes AT 12 months and outcomes OVER 12 months. I assumed the latter was the difference from the baseline to the 12 month point, whereas the former was the raw comparison at 12 months, but this may be incorrect. Hence, it might be helpful to make this more explicit at some point in the paper, or use a more direct/explicit language (e.g., 'Change from baseline to 12 month assessment'). 

p12, line 72. It didn't seem correct to claim that the evidence was for a sustained effect 'during' the 12 month period, as very little was known from the 12 week test to the 12 month endpoint test. 'Over' would seem a more accurate description of the sustained effect. 

The per-student cost is made more salient in the paper, and is the cost provided in the abstract, whereas the per-case cost seems to me a more common and meaningful indicator of the actual economic cost of the intervention. To say that it costs $4 per school pupil to provide this intervention, when most of these pupils are not, in fact, receiving the intervention, does not feel particularly meaningful or precise to me. 

p.5, line 89. I was not familiar with term 'retrospective ingredients' approach. If this section is intended to be accessible to a non-specialist readership it may be advisable to use a more accessible or clearly descriptive term. 

p9, line 166. It wasn't clear whose assent is being referred to here: adolescents or parents/carers. 

p12, line 198. Where was the reason for noncompletion from? Was this self-report or otherwise? 

p13, Although details of the primary outcome measures are given in the original paper, I think it would be helpful to describe just a bit more here, particularly as the format of the YTP is somewhat non-standard/idiographic. 

p14, Can a little more be given about the nature of the self-report measure of PS materials and skills: how were the questions phrased here? 

p20, I wonder if there is a need for the detail on the primary caregivers' education and occupation, or whether this might go into an appendices to shorten the paper somewhat. 

p23, line 339-343: again, I struggled somewhat to understand what the statistic being given here was, as compared with the comparison of outcomes at 12 months. Is this the change from baseline? What does it mean 'SDQ = -1.23'? (I wonder if this is a typo and should read: 'SDQ: AMD ='

In the Discussion, I think more could have been said of the fact that neither the use of the booklet, or of problem solving skills, significantly mediated the outcomes. It states on page 31, line 427, that the findings strengthen the evidence base for the effects of problem solving, but without any evidence of a mediating effect, and given that the comparison condition was also problem solving (without counsellor input), it might be argued that the findings do not particular support the value of problem solving practices, per se but rather contact with a counsellor (for instance, as a consequence of relational common factors). Having said that, it is mentioned that the power was not sufficient for this mediation analysis and, if it is indeed the case that nothing can be gleaned from this null finding due to under-powering, I think it would be important to state that earlier on. 

S8 Table. It might be the formatting, but I struggled to understand what the means were that were being presented here. Is the data definitely in the right boxes?

Reviewer #2: The study by Malik et al. provides evidence to address two substantial gaps in the literature - the sustained impact and cost of brief mental health interventions in LMIC adolescents. Among the strengths are the use of a "real-world" population and rigorous statistical assessment of outcomes and potential mediators and moderators. While the effect on mental health symptoms demonstrated in the manuscript is small, the authors include a fair reporting of these modest results and a clear discussion of important limitations. My concerns and suggestions to the authors are largely minor issues and points of clarification, outlined below.

Methods

Line 226-227: Are SDQ scores of 19 and 20 considered the clinical threshold here, as they were considered the cut-off scores for inclusion?

Lines 254-255 are not clear. The sentence describes adjustment for baseline measure, school, and missing FU, but the parenthetic (which seems to specify the particular variables adjusted for) includes age, class, and week of enrollment and do not include not missingness.

Discussion

How much variability is expected in cost of the intervention through other implementers? It would seem that other NGOs or governmental bodies would be needed to provide scale-up, at least beyond the 20 schools modeled. It would be helpful in considering the larger-scale impact if the authors could comment on how costs may vary in the paragraph starting at line 451.

Reviewer #3: This is an interesting RCT on the effectiveness and costs of a lay counsellor-delivered, brief problem-solving intervention for adolescent mental health problems in urban, low-income schools in

India. However, there are quite a few major issues needing attention.

1. Sample size. There is no sample size calculation in the paper which is inadequate. The original trial was designed for 6 and 12 weeks outcomes but modified to 12 months, why? and on what bases? However, as written in the paper "the sample size calculation did not take into account the numbers expected for the 12-month assessments", it means the current trial was not properly powered so would not be able to answer the research questions with confidence.

2. Loss of follow-ups. About 30% of loss of follow-ups is of a big concern. Not matter how data was imputed and sensitivity analyses were done, the reliability and robustness of the results are subject to scrutiny.

3. Adherence to intervention. In the intervention arm, there are 97 completed & 28 discontinued intervention, therefore only 97 are per protocol with around 22% of those in the treatment arm didn't complete the intervention. However, this issue was never discussed in the paper or addressed in the analyses.

4. Statistical analyses. The main results were shown in table 2 but very confusing. As shown in table 2, the control arm has 95 and intervention arm has 81 participants which shows this is complete case analysis rather than the 'intention to treat' analysis claimed by the authors. Also the section on 'over 12 months' with 339 controls and 325 interventions is very confusing as it's never specified in the stats analysis section or anywhere. It should be outcomes at 12 months only in table 2.

5. Interpretation of results. What we can see from table 2 main analysis on the 12-month outcomes and also sensitivity analysis from the S3 table, there were either no differences or borderline differences in both primary or secondary outcomes between intervention and control arms. Coupled with inadequate sample size, 30% loss of follow-ups and 22% non-adherence to protocol (intervention arm), there is no evidence with sufficient power to support authors' claim on the usefulness of the intervention.

6. Table 1 should be on comparing two arms rather than on loss of follow-ups, and it's also not properly done. For categorical variables, the percentage should be done column-wise rather that row-wise so that can make sense. 

7. Not able to comment on the costing models as need specialist on this part.

[LINK]

---

## [Decision Letter · Decision Letter 2]

4 Jun 2021

Dear Dr. Patel,

Thank you very much for submitting your revised manuscript "Sustained effectiveness and costs of a lay counsellor-delivered, brief problem-solving intervention for adolescent mental health problems in urban, low-income schools in India: 12-month outcomes of a randomized control trial" (PMEDICINE-D-20-05026R2) for consideration in PLOS Medicine’s Special Issue: Global Child Health: From Birth to Adolescence and Beyond. 

Your paper was evaluated by a senior editor and discussed among all the editors here. It was also discussed with an academic editor with relevant expertise, and sent to the three original reviewers, including a statistical reviewer. The reviews are appended at the bottom of this email and any accompanying reviewer attachments can be seen via the link below:

[LINK]

In light of the remaining points raised by the reviewers, I am afraid that we will not be able to accept the manuscript for publication in the journal in its current form, but we would like to consider a revised version that addresses the reviewers' and editors' comments. Obviously we cannot make any decision about publication until we have seen the revised manuscript and your response, and we plan to seek re-review by one or more of the reviewers. 

We expect to receive your revised manuscript by Jun 25 2021 11:59PM. Please email us (plosmedicine@plos.org) if you have any questions or concerns.

We look forward to receiving your revised manuscript. 

Sincerely,

Caitlin Moyer, Ph.D.

Associate Editor 

PLOS Medicine

plosmedicine.org

1. How did the authors assess fidelity of lay counsellors to the intervention? This is important if lay counsellors are used and scale-up is recommended in a context where mental health resources for adolescents are limited and lay counsellors are needed for program delivery.

2. Please address the remaining points of the reviewers. Given Reviewer 3’s remaining concerns with the limitations of the study, please acknowledge these limitations as thoroughly as possible, and temper or qualify claims (in the Abstract and Discussion) in light of these limitations, where appropriate.

3. Response to reviewer comments point # 41: It would be helpful to have the table of baseline demographic and clinical characteristics for the two arms. Thank you for noting that this information for the two arms was presented in the primary trial paper (Michelson et al, 2020). Although it can be found elsewhere, it would be helpful to include here as well for ease of access.

4. Data availability statement: Thank you for providing link for accessing the de-identified data. If possible, please provide additional information, or a more direct link, for accessing the specific data included in this study.

5. Abstract Line 55-56: If helpful, you could clarify here where you describe that “A protocol modification was made…” that “the trial steering committee recommended a post-hoc extension of the follow-up period to 12 months” or similar language (to provide some context for the 12 month outcomes).

6. Abstract: Methods and Findings: Please include a sentence summarizing adverse events, if relevant. In the last sentence of the Abstract Methods and Findings section, please describe the main limitation(s) of the study's methodology.

7. Author summary: Why was this study done? For this section, it may be helpful to include an introductory point on the broader context for the study (beyond mentioning the previous report on the trial findings).

8. Results: Line 401-402: Please present the results in the text for the intervention effect on SDQ externalization symptom scores over 12 months, although you note this did not reach statistical significance.

9. Results: Line 404: We suggest revising to clarify that while some of the effects for SDQ total difficulties no longer reached statistical significance in the sensitivity analyses, the effect did not appear to be attenuated for the sensitivity analysis adjusted for missing data over 12 months.

10. References: Please ensure the use of the "Vancouver" style for reference formatting (including Journal title abbreviations), and see our website for other reference guidelines https://journals.plos.org/plosmedicine/s/submission-guidelines#loc-references

11. Figure 2a and 2b: Please mention in the legend that the error bars represent mean with SD, if accurate. Please also fully define abbreviations YTP and SDQ in the legend.

12. Table 2: Please note if values are reported as mean score with (SD), if accurate.

13. CONSORT checklist: Thank you for including the CONSORT checklist. Please remove references to page numbers, and instead use sections and paragraph numbers to refer to locations within the text. For “Funding” we suggest reporting the location as “Financial Disclosure statement” or similar.

Comments from the reviewers:

Reviewer #1: Thank you for the opportunity to review the revised submission for this important study, showing the longer term effects of a school-based problem solving intervention. I believe this is an important and robustly conducted and analysed study that is close to completion for publication. 

I appreciated the authors' responses to the reviewers comments and their willingness to engage with a range of points and issues. As Reviewer 1, I have a few final recommendations for reconsideration by the authors, and one or two additional points. 

Point 21: I continue to feel that the use of the term 'over' 12 months, as opposed to 'at' 12 months, is not as clearly defined as it could be across the MS. It may be that this is common terminology that I am not familiar with and, if so, I am happy to leave this point; but as an informed reader I was still left a bit uncertain what this meant particularly, for instance, on the data presented in Table 2. I think I now understand that the 'over 12 months' score is the predicted score based on the three timepoint, assuming linear change (as opposed to the exact score at 12 months). In the abstract this is explained as, 'the latter assumed a constant effect across three follow-up points'. That gets me closer to what is being described here, but it still doesn't quite tell me WHAT the score is (for instance, is it the predicted 12 months score based on the assumption of a constant effect, or the change over time assuming a constant effect?). From Table 2 it seemed to be the former, but 'over' just doesn't feel like the right word here. 'Predicted' or 'estimated' might be clearer? 

Point 23: Although I can see the authors' point here, there seems something of an incongruity to me to be presenting all clinical change (in the abstract) on a per-case basis, but then the costs on a per-student basis. If the principal concern were to look at the whole school effects, surely the mental wellbeing outcomes should be per student as well. Perhaps a 'compromise' here would be to provide the per participant costs along with the per-student costs in the abstract: for instance in parenthesis after the USD4. 

Line 209: It's just a little confusing that it talks about the people who delivered the intervention, and then subsequently the counsellors. I'm 99% certain, as a reader, that this is the same group, but perhaps just change 'The counsellors' to 'These counsellors' on line 201-211 to make clear it's the same people. 

Line 245: It's good to have some explanation of why there are two primary outcomes. However, it does then beg the question of whether you looked to see any cross-lagged effects from changes in prioritized problems to effects on mental health symptoms. I couldn't see any of that analysis here. Is it something to suggest for future research? I guess that depends on how strongly you hold this hypothesis. If just a loose guess, it might make more sense on lines 244-246 to talk about these as two important outcomes rather than suggesting a causal relationship. 

Table 2: Apologies if I missed this, but I wasn't sure why you had the 'at' and 'over' AMDs/ESs/ps for the two primaries, but then not for the secondaries. Do these need adding in too? If not, are the data given 'at' or 'over'? 

Line 429 says that the per-case cost was USD 158, but this seems to reduce to USD18 at line 446. Is a 'per-student treated' cost the same as a 'per-case' cost (if so, please align terminology and, if not, please explain the former term). More importantly, I did struggle to understand how the cost when the intervention was scaled up was reduced by almost 90%. Are those calculations definitely right? (I checked S3 but that didn't really help explain the magnitude of reduction). Perhaps a few added words in the text here might help the reader to understand why the cost becomes so much less. 

Table 3: The per-treated case cost is now 23 USD. Should that be 18, as above. 

I really failed to understand the table here. Is there some way it can be made clearer? How is it that the Total cost in the top row is not the total of the subsequent costs? Why is the per-school cost greater in the scale up version than for the six schools, but the pre-treated case cost so much lower? As a reader (albeit not expert in economic analysis), this table, and the summary of it (lines 434-446) could do with some further work to really bring out the key findings and the logic of the analysis.

lines 559-562: It seemed strange to end the paper by referencing econd line interventions. No evidence was presented to support the claim that this would improve effects, and it seemed to take away from the value of the first-line intervention effects being reported here. 

Reviewer #2: I am satisfied with the authors' responses to my initial comments. However, their clarification on the SDQ clinical threshold used and additional revisions regarding other reviewer comments have raised some additional major concerns.

Major Concerns:

I am either confused by the presentation or there is an error in the data entered in Table S8. How did completers have a baseline SDQ mean of 8? Wouldn't it have had to be higher than 19, considering that was the cutoff for inclusion? Is baseline at the end of the treatment and not the beginning of the trial? It looks like something may be wrong with the YTP cells as well, with completers having a mean score of 1.5 at baseline and 2.4 at follow-up while non-completers have baseline of 7.1 (much closer to the mean baseline YTP score in Table 1). Neither of these low baseline scores seem reasonable when looking at Figure 2 graph means and error bars. If these numbers are in fact correct, the authors need to clarify how the SDQ baseline mean was 8 (far below the inclusion cutoff) and why completers' baseline scores for both the SDQ and YTP were so low compared to non-completers. If they are incorrect, in addition to updating results text and the table, the authors need to change the discussion paragraph on dose-response to reflect the correct finding.

I think there is not enough attention paid to the fact that the average symptom reduction over time and the percent of remitters in the control and intervention are quite similar. Yes, there is a significantly higher reduction in the intervention, but the AMD for the SDQ total difficulties is <2. Considering 1) the SDQ total scores can range from 0-40, 2) 19/20 is the clinical cutoff, and 3) both the intervention arm and control arm seem to have an average symptom reduction of ~10 points over the 12 months (Table 2B), I think the authors need to comment on the clinical significance of such a small difference between arms. Moreover, while I appreciate the authors edits to remove emphasis on problem-solving and clarify the potential optimization by non-specialist providers as well as their mention that the study was not powered to detect mediators (lines 467-479), I think this needs to be more hashed out. Are the problem-solving booklets alone a good intervention? Considering the majority of costs came from personnel, the control condition may actually be a more cost-effective intervention. Is it possible that there was contamination from adolescents in the intervention group discussing their treatment with peers in the control group? I think there also may be a typo in lines 416-418 "There was no evidence that perceived stress, use of booklets of problem-solving skills at the 12-week endpoint mediated the intervention effect on primary outcomes at 12 months (S6-S7 Tables)."

Minor Comment

In sensitivity analyses adjusting for missingness, it appears that the SDQ impact score was not significantly different in the intervention vs. control over 12 months (Table S3). The authors should add to this sentence in addition to the mention of the weakened effect seen for the SDQ total difficulties score in the results and discussion sections.

Reviewer #3: Thanks authors for their effort to improve the manusript. However, I am not satisfied with the response and revision at all. The authors failed to address almost every single one of my comments/concerns. Just to reiterate,

there were either no differences or borderline differences in both primary or secondary outcomes between intervention and control arms. Coupled with inadequate sample size (not powered at all), 30% loss of follow-ups and 22% non-adherence to protocol (intervention arm), there is no evidence with sufficient power to support authors' claim on the usefulness of the intervention.

Key points:

1) The analyses are not 'intention to treat' at all. Please stop using it and acknowledge as a limitation carefully.

2) The study is not powered for the 12-month primary outcome at all. It looks more like an observational study with follow-ups.

3) For RCT, it's not acceptable without a baseline table 1.

[LINK]

---

## [Decision Letter · Decision Letter 3]

13 Aug 2021

Dear Dr. Patel,

Thank you very much for re-submitting your manuscript "Sustained effectiveness and costs of a lay counsellor-delivered, brief problem-solving intervention for adolescent mental health problems in urban, low-income schools in India: 12-month outcomes of a randomized controlled trial" (PMEDICINE-D-20-05026R3) for consideration in PLOS Medicine’s Special Issue: Global Child Health: From Birth to Adolescence and Beyond.

I have discussed the revised paper with my colleagues and the Special Issue Guest Editors, and it was also seen again by three reviewers. I am pleased to say that provided the remaining editorial and production issues are dealt with we are planning to accept the paper for publication in the journal.

[LINK]

We look forward to receiving the revised manuscript by Aug 19 2021 11:59PM.   

Sincerely,

Caitlin Moyer, Ph.D.

Associate Editor 

PLOS Medicine

plosmedicine.org

Requests from Editors:

1. Title: We suggest revising the title to: “Effectiveness and costs associated with a lay counsellor delivered problem-solving mental health intervention for adolescents in urban, low-income schools in India: 12 month outcomes of a randomized controlled trial.” or please revise similarly with the aim of making the title as concise as possible.

2. Data availability statement: Thank you for providing a link to access the de-identified data. We request that you provide more specific information to facilitate access to the data. Please provide a more direct link to the data, and/or include the DOI or accession number for the dataset.

3. Response to reviewer 1: Please address the comment of the reviewer regarding the mediation analysis and lack of power. 

4. Response to reviewer 3: Please balance the description of results and interpretation in response to Reviewer 3’s points regarding the loss to follow up at 12 months and the borderline significance of some findings (namely the SDQ Total Difficulties Scale comparison at 12 months, taken together with the results from the sensitivity analysis). 

 The editors ask that acknowledgement of this limitation be given more explicit emphasis throughout where implications of the results are considered, particularly in the Abstract and Discussion.

5. Abstract: Line 74: “The intervention effect was stable for most outcomes on sensitivity analyses; however, for SDQ Total Difficulties and impairment, the effect was slightly attenuated.” Please clarify the sensitivity analyses mentioned here (that this was the sensitivity analysis adjusting for missing data) rather than just saying the sensitivity analysis showed an attenuated effect.

6. Abstract: As the last sentence of the Abstract Methods and Findings section, please describe the main limitation(s) of the study's methodology.

7. Abstract: Line 82-83: In line with reviewer 3’s comments, we think it may be helpful to provide slightly more general detail on why caution is needed given the post-hoc extension of the trial. We suggest also mentioning the specific limitations as the last sentence of the Methods and Findings section of the abstract.

8. Discussion: Line 487-488: Referring back to Reviewer 3’s point, it may be helpful here to qualify the conclusions slightly, noting that the evidence supports an effect 12 months, but the effects are small particularly for SDQ Total Difficulties.

9. Tables: For the notes in the legend, we suggest not numbering the notes across tables, but instead numbering the notes beginning with “1” for each table.

10. Table 3: Please provide the definition for abbreviation (USD) in the legend.

11. References: Please use the "Vancouver" style for reference formatting, and see our website for other reference guidelines: https://journals.plos.org/plosmedicine/s/submission-guidelines#loc-references

Please double check formatting of all references, including journal title abbreviations. For example, in reference 2, The Lancet should be Lancet. In reference 5, Behaviour Research and Therapy should be Behav Res Ther. In reference 7, Lancet Child and Adolescent Health should be Lancet Child Adolesc Health. Please revise throughout the list.

12. Figure 1 Title and S2 Appendix: In the title of Figure 1 and for the title of S2_Appendix listed on page 41, please capitalize CONSORT.

13. S8 Table: In the file and title name for this table listed in the main text (page 41), please change “compliers” and “non-compliers” to completers. Thank you for making this change within the table file.

14. S3 Appendix: Please provide a clean version of the file (there is some mark-up in the reference to S1a, S1 b Tables). Please format the reference in this document similar to those in the main text (with the in text citation within square brackets rather than superscript, and the appropriate reference formatting). 

Comments from Reviewers:

Reviewer #1: Thank you for the further work on this manuscript and the detailed response. It makes an excellent and much-needed contribution to our understanding of effective intervention for adolescent mental health. Congratulations on your work and your encouraging findings. 

I have just a few final comments: 

Abstract: The actual per-case (and per-student) costs need to be given in the abstract, as this is the principal finding from the study (USD158). If the estimated/scaled up per case/per-student costs are also given in the abstract (USD4/23), as present, it should be made clear that this is an 'estimate'. 

line 512-514. It needs to be made clearer here that the mediation effects of specific mechanisms/processes were tested, and not found to significantly mediate. This seems to be the first time that the lack of power for this analysis is given, and it would be helpful if this was stated earlier in the methods. An indication of the specific power for this analysis may be helpful if possible. 

Reviewer #2: I am satisfied with the authors' response and most recent revisions.

Reviewer #3: Thanks authors for their effort to improve the manuscript. The authors have addressed some of my concerns such as clarifying the 'intention to treat' and also adding the baseline Table 1. However, I am still not convinced/satisfied with the response to point 29 on sample size, power and statistical significance. 

1) The authors said "Regarding the point about difference between arms and inadequate sample size: Table 2 (page 23-24) shows that there were significant intervention effects (p<0.05) on all primary and secondary outcomes except for the SDQ Externalizing subscale score (p=0.13) and SWEMWBS score (p=0.06)". However, out of 9 comparisons in table 2, there are 7 p-values ranging from 0.02 to 0.06 therefore means borderline significane (very close to 0.05) in 78% of all the results. Also the sensitivity analyses with missing data imputation didn't help. Basically the results in table 2 showed very weak and inconclusive evidence (if any) on the effectiveness of the intervention, which has never been acknowledged and discussed in the paper. 

2) The authors said "we respectfully disagree that the study is not powered for the 12-month primary outcomes". However, it says in page 10 "The estimated sample size for the originally stipulated primary endpoint assumed loss to follow-up of 15% at six weeks (based on piloting) and a 1:1 allocation ratio", therefore clearly the sample size was calculated for outcome at 6 weeks rather than at 12 months. Now the primary analyses have changed to 12 months but the sample size was still from that at 6 weeks? Just to re-iterate, therefore, the study was not powered (sample size) for primary outcomes at 12 months. The sample size calculation at 6 weeks doesn't match the primary outcomes at 12 months. Due to the inadequte sample size, mostly borderline results and the 30% loss of follow-ups and 22% non-adherence to protocol (intervention arm), the reliability and believability of the results are subject to scrutiny.

[LINK]

---

## [Editor Report · Decision Letter 4]

20 Aug 2021

Dear Dr Patel, 

On behalf of my colleagues and the Academic Editor, Kathryn Yount, I am pleased to inform you that we have agreed to publish your manuscript "Effectiveness and costs associated with a lay counsellor delivered, brief problem-solving mental health intervention for adolescents in urban, low-income schools in India: 12-month outcomes of a randomized controlled trial" (PMEDICINE-D-20-05026R4) in PLOS Medicine’s Special Issue: Global Child Health: From Birth to Adolescence and Beyond.

PRESS

Sincerely, 

Caitlin Moyer, Ph.D. 

Associate Editor 

PLOS Medicine